# Soil Candidate Phyla Radiation Bacteria Encode Components of Aerobic Metabolism and Co-occur with Nanoarchaea in the Rare Biosphere of Rhizosphere Grassland Communities

Alexa M. Nicolas,[a] Alexander L. Jaffe,[a] Erin E. Nuccio,[b] Michiko E. Taga,[a] Mary K. Firestone,[c,d] Jillian F. Banfield[c,e,f,g]

[a]Department of Plant and Microbial Biology, University of California, Berkeley, Berkeley, California, USA

[b]Nuclear and Chemical Sciences Division, Lawrence Livermore National Laboratory, Livermore, California, USA

[c]Department of Environmental Science, Policy, and Management, University of California, Berkeley, Berkeley, California, USA

[d]Earth and Environmental Sciences, Lawrence Berkeley National Laboratory, Berkeley, California, USA

[e]Department of Earth and Planetary Science, University of California, Berkeley, Berkeley, California, USA

[f]Chan Zuckerberg Biohub, San Francisco, California, USA

[g]Innovative Genomics Institute, University of California, Berkeley, Berkeley, California, USA

**ABSTRACT** Candidate Phyla Radiation (CPR) bacteria and nanoarchaea populate most ecosystems but are rarely detected in soil. We concentrated particles of less than $0.2\,\mu$m in size from grassland soil, enabling targeted metagenomic analysis of these organisms, which are almost totally unexplored in largely oxic environments such as soil. We recovered a diversity of CPR bacterial and some archaeal sequences but no sequences from other cellular organisms. The sampled sequences include Doudnabacteria (SM2F11) and Pacearchaeota, organisms rarely reported in soil, as well as Saccharibacteria, Parcubacteria, and Microgenomates. CPR and archaea of the phyla Diapherotrites, Parvarchaeota, Aenigmarchaeota, Nanoarchaeota, and Nanohaloarchaeota (DPANN) were enriched 100- to 1,000-fold compared to that in bulk soil, in which we estimate each of these organisms comprises approximately 1 to 100 cells per gram of soil. Like most CPR and DPANN sequenced to date, we predict these microorganisms live symbiotic anaerobic lifestyles. However, Saccharibacteria, Parcubacteria, and Doudnabacteria genomes sampled here also harbor ubiquinol oxidase operons that may have been acquired from other bacteria, likely during adaptation to aerobic soil environments. We conclude that CPR bacteria and DPANN archaea are part of the rare soil biosphere and harbor unique metabolic platforms that potentially evolved to live symbiotically under relatively oxic conditions.

**IMPORTANCE** Here, we investigated overlooked microbes in soil, Candidate Phyla Radiation (CPR) bacteria and Diapherotrites, Parvarchaeota, Aenigmarchaeota, Nanoarchaeota, and Nanohaloarchaeota (DPANN) archaea, by size fractionating small particles from soil, an approach typically used for the recovery of viral metagenomes. Concentration of these small cells ($<0.2\,\mu$m) allowed us to identify these organisms as part of the rare soil biosphere and to sample genomes that were absent from non-size-fractionated metagenomes. We found that some of these predicted symbionts, which have been largely studied in anaerobic systems, have acquired aerobic capacity via lateral transfer that may enable adaptation to oxic soil environments. We estimate that there are approximately 1 to 100 cells of each of these lineages per gram of soil, highlighting that the approach provides a window into the rare soil biosphere and its associated genetic potential.

**KEYWORDS** CPR, DPANN, archaea, bacteria, environmental microbiology, metagenomics, soil, soil microbiology, symbiosis

Address correspondence to Jillian F. Banfield, jbanfield@berkeley.edu.

nteractions among soil microorganisms impact biogeochemical cycling and overall ecosystem function. A recent metagenomic analysis of soil microbial communities revealed that many steps of key reaction pathways central to transformations in soil are partitioned among coexisting organisms (1). Other interactions are mediated by molecules such as vitamins and antimicrobial compounds (2–4). Furthermore, there is the potential for a variety of symbiotic interactions, including those that involve obligate reliance on coexisting organisms, for even the most basic requirements (5, 6). Candidate Phyla Radiation (CPR) bacteria and DPANN archaea (an acronym of the names of the first included phyla: Diapherotrites, Parvarchaeota, Aenigmarchaeota, Nanoarchaeota, and Nanohaloarchaeota) are detected across ecosystems and are often predicted to be obligate anaerobic (epi)symbionts that depend on other organisms for basic cellular building blocks (5, 7, 8). However, CPR bacteria and DPANN archaea have rarely been studied in relatively oxic environments or identified in soil (1, 9, 10). Genome-resolved metagenomic analyses circumvent the limitations of isolation-based methods that fail for organisms unable to grow alone and for bacteria that evade detection by primers used in 16S rRNA gene surveys (11); yet, there are few reports of CPR metagenome-assembled genomes (MAGs) from soil (12–15) almost certainly because of the rarity of these bacteria.

Prior studies of groundwater have taken advantage of the observation that CPR bacteria have ultrasmall cells that pass through 0.2-$\mu$m filters and enable genome recovery for these organisms (11). Studies of other systems reveal that size fractionation of particles prior to sequencing impacts the composition and function of metagenomes (16). However, to our knowledge, studies of 0.2-$\mu$m filtrates from soil have focused on viromes and have not assessed their microbial contents (17–19). Here, we took advantage of the expected very small sizes of CPR bacteria and DPANN archaeal cells (20) to concentrate them from soil. Thus, we could test the hypothesis that these anaerobic organisms are understudied parts of the rare soil biosphere, where they may have evolved pathways to persist in relatively oxic environments. We sequenced concentrated soil effluent that had passed through a 0.2-$\mu$m filter used to remove larger cells and recovered a diversity of bacterial and archaeal sequences.

We sampled rhizosphere-associated soil from the top 10 cm of an annual grassland from the Hopland Research and Extension Center in February 2018. For a subset of the soil samples, we added a potassium citrate-based buffer and collected the effluent, which was passed through a 0.2-$\mu$m filter, concentrated, and treated with DNase to remove extracellular DNA that could have derived from larger lysed cells (Fig. 1a; see also Text S1 in the supplemental material) (21, 22). To evaluate enrichment, bulk DNA was extracted from the same soil samples for whole-community shotgun DNA sequencing, generating what are here referred to as "bulk metagenomes." Approximately 20 Gbp of sequence was obtained from each of six concentrates and two bulk samples. In addition to recovering viral sequences and mobile elements from these small-particle-concentrate metagenomes, we reconstructed sequences from CPR and nanoarchaeal genomes. From these data, we resolved 26 draft genomes that were >70% complete (estimated using a CPR-specific single copy gene set [11]), with <10% contamination derived from either CPR or DPANN. No CPR or DPANN genomes were recovered from the bulk metagenomes.

Sequences from cells of <0.2 $\mu$m in size were almost exclusively from 15 lineages of CPR bacteria and one DPANN archaeal phylum (Fig. 2; see also Fig. S1). Importantly, CPR and DPANN sequences were completely absent in bulk metagenome samples and were only detectable at very low, if any, coverage via read mapping to assembled sequences from the small-size-fraction metagenomes (Fig. 1c). Furthermore, from the 74 bacterial 16S rRNA sequences recovered from the concentrate metagenomes, all of which were assigned to CPR lineages, we predict that more than half (42 16S rRNA gene sequences) would not have been detected using standard amplicon sequencing primers. Notable was the phylum-level diversity of CPR lineages in the concentrate metagenomes. Previously, a genome of TM7 (Saccharibacteria) was reported

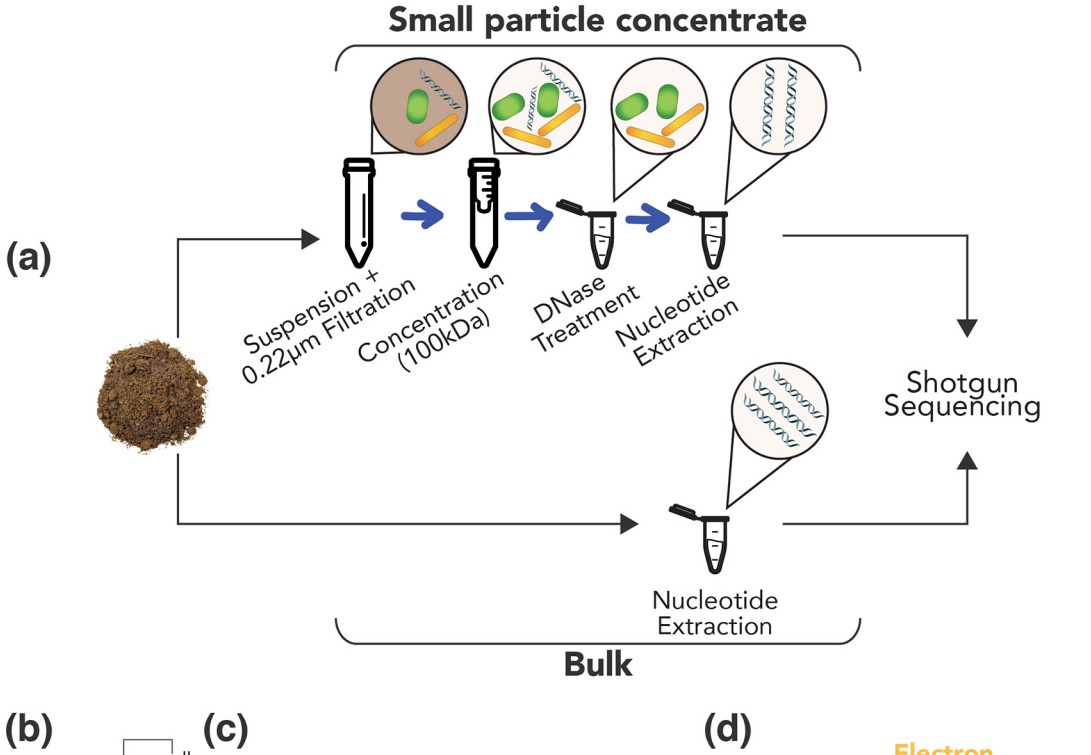

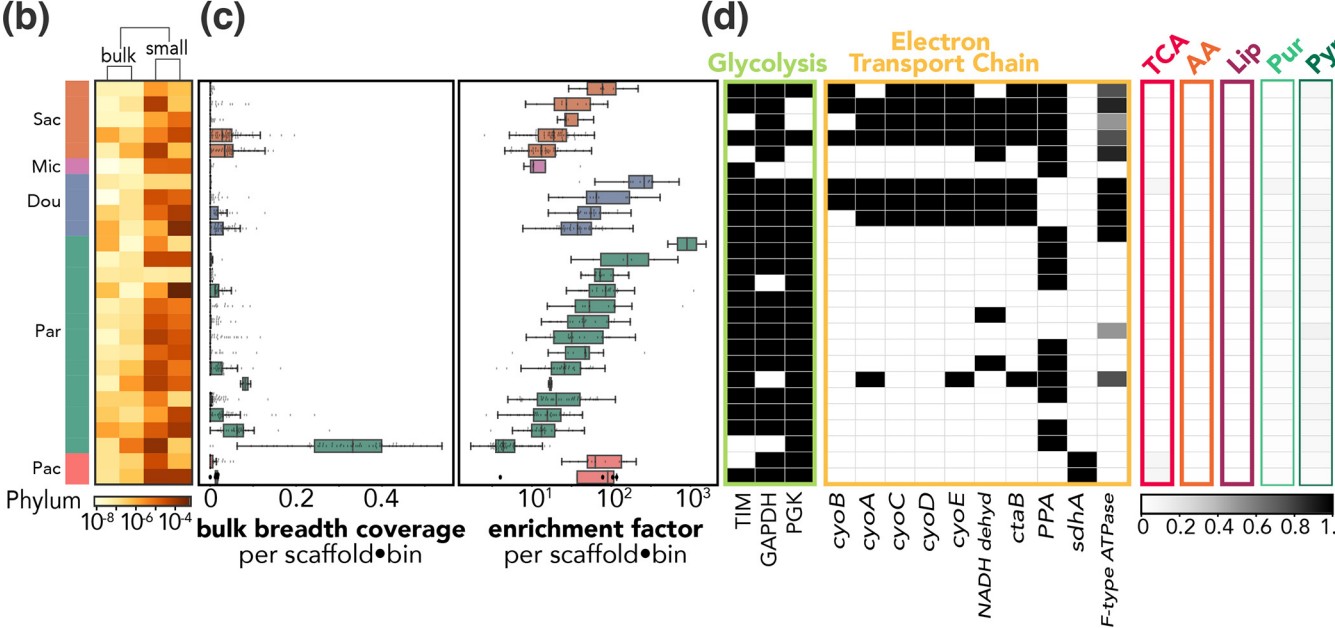

**FIG 1** Enrichment and metabolic profiles of CPR bacteria in soil concentrate metagenomes. (a) Method for concentration of small particles from soil for metagenomic sequencing (top) compared to sample preparation methods for bulk soil metagenomes (bottom). (b) Heat map showing relative abundance of 26 organisms by phylum across bulk metagenomes and concentrate metagenomes. Sac, Saccharibacteria; Mic, Microgenomates; Dou, Doudnabacteria; Par, Parcubacteria; Pac, Pacearchaeota. (c) Coverage-based metrics showing recovery and enrichment in all concentrates combined relative to that in bulk fractions, combined as boxplots. (Left) Breadth of coverage of scaffolds comprising each genome (bin) in the bulk fraction. (Right) Enrichment factor (i.e., relative abundance of a scaffold from the concentrate metagenome over a scaffold's bulk metagenome relative abundance) for each genome. (d) Metabolic analysis of each genome, including (i) presence of each of three glycolysis genes that are highly conserved among CPR bacteria, (ii) genes involved in the electron transport chain (NADH dehydrogenase; *ctaB*, heme O synthase [EC 2.5.1.141]; *PPA*, inorganic pyrophosphatase [EC 3.6.1.1]; *sdhA*, succinate dehydrogenase/fumarate reductase, flavoprotein subunit [EC 1.3.5.1 1.3.5.4]), and (iii) percentage completeness (grayscale) of F-type ATPase, the TCA cycle (tricarboxylic acid cycle), and pathways for amino acid biosynthesis (AA), lipid biosynthesis (Lip), purine biosynthesis (Pur), and pyrimidine biosynthesis (Pyr).

from the same soil but sampled at less than $1\times$ coverage from bulk soil (6), and Microgenomates and Parcubacteria have been genomically sampled at low abundance (13, 23). To our knowledge, this is one of very few reports of Pacearchaeota and Doudnabacteria (14, 15) in soil and the first report of a novel clade of Saccharibacteria.

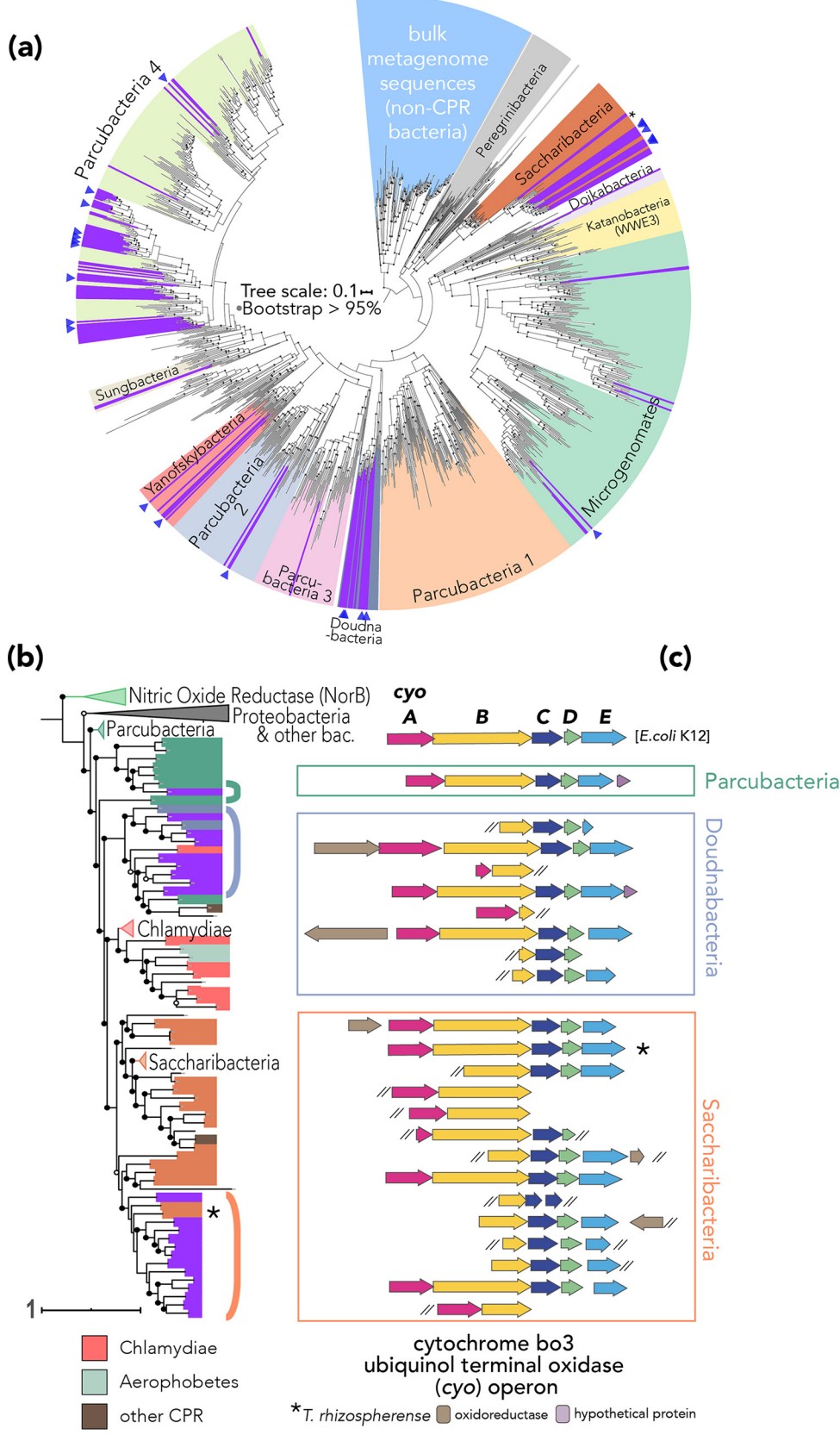

**FIG 2** Soil CPR phylogeny and cytochrome operon synteny. Sequences assembled from the small concentrate metagenomes are shown in purple. (a) RpS3 tree of CPR bacteria rooted using RpS3 sequences that were

mSystems®

Comparing sequence coverage from concentrates to that from the bulk metagenomes, we calculate that filtration enriched the relative abundance of genomes by 100 to $1,000\times$ (Fig. 1c). We approximate, given the relative abundances of the most- and least-abundant CPR genomes in each of the bulk and concentrate metagenomes, that CPR cells may comprise on the order of 1 to 100 cells per gram of soil. Given estimates of $10^9$ microbial cells per gram of soil, this would equate to, at maximum, $\sim10^{-5}\%$ of microbial cells in a gram of soil (Text S1).

Given the unique challenges of the soil environment for microbes, we next assessed whether these soil CPR and DPANN organisms exhibited similar traits to those of their counterparts in other environments. Recent studies show that CPR bacteria generally appear to have the capacity for glycolysis and fermentation (24) but often lack complete pathways to synthesize nucleotides *de novo* and have many gaps in metabolism that suggest an obligate symbiotic lifestyle (8). We find that most of the genomes from this sampling effort encode the three central glycolysis enzymes reportedly found in nearly all CPR bacteria: triose phosphate isomerase (TIM), glyceraldehyde 3-phosphate (GAPDH), and phosphoglycerate kinase (PGK) (24). The genomes also contain few if any tricarboxylic acid (TCA) cycle genes and lack the vast majority of genes of the electron transport chain and for synthesis of lipids and nucleotides (Fig. 1d), suggesting they live anaerobic lifestyles and depend on resources from other organisms (8). However, we identified an operon encoding a multisubunit cytochrome $bo_3$ ubiquinol terminal oxidase in three Doudnabacteria genomes, eight Saccharibacteria genomes, and one Parcubacteria sequence as well as in unbinned CPR phylum sequences from the concentrate metagenomes. We then performed a synteny analysis (Fig. 2c) to compare these loci to a related one from the first Saccharibacteria genome described from soil, *Candidatus* Teamsevenus rhizospherense (6). The comparison shows a gene order for the *cyo* operon identical to that in the highly studied *Escherichia coli* K-12 operon (25) and in the *T. rhizospherense* genome (6), although some CPR loci were incomplete due to assembly fragmentation. Several CPR loci also included an open reading frame (ORF) annotated as an oxidoreductase or a conserved hypothetical protein. While the genomes recovered do not contain quinone biosynthesis genes, the combination of this ubiquinol oxidase and the associated oxidoreductase (Fig. 2c), which often co-occur in genomes encoding an NADH dehydrogenase and, to varied completeness, F-type ATPase (Fig. 1d), suggests the possibility of some aerobic respiratory capacity. Perhaps, some form of aerobic respiration may be common in soil-associated Saccharibacteria specifically and perhaps in soil CPR more broadly. We thus hypothesize that this operon may confer an adaptive advantage for CPR bacteria to live in aerophilic environments such as surface soil.

Next, we generated a maximum-likelihood tree of subunit 1 (CyoB) of the *cyo* operon to test whether the operon exhibited a pattern of vertical inheritance in our CPR genomes (Text S1; Fig. 2b). This analysis suggests that this gene cluster has been laterally transferred from other bacteria, such as *Proteobacteria* or *Chlamydiae*, into these CPR bacteria at least once, with perhaps different origins for gene clusters in Parcubacteria and Doudnabacteria from those in Saccharibacteria. Furthermore, based on CyoB phylogeny, the sequences from *T. rhizospherense* appear more closely related to Saccharibacteria sequences from this study than to the RpS3 phylogeny, which may further underscore local adaptation to soil.

Here, we conducted a targeted study of CPR bacteria and nanoarchaea in a soil ecosystem to expand our understanding of rare soil-dwelling microbes. Using typical sequencing

**FIG 2** Legend (Continued)

assembled from bulk metagenomes, in light blue. Blue triangles denote draft genome recovered. Nodes with bootstrap values greater than or equal to 0.95 are marked as filled black circles. (b) Phylogenetic relationships of cytochrome $bo_3$ ubiquinol terminal oxidase subunit I across bacterial phyla. Circles overlaid on nodes correspond to support values (unfilled, $>0.50$; filled, $>0.70$). *, the placement of *T. rhizospherense* CyoB (b) and its operon in (c) (6). Brackets next to tree tips correspond to phyla by color (green, Parcubacteria; blue, Doudnabacteria; orange, Saccharibacteria) and to sequence order in the synteny diagram (c). Tree rooted using a heme-copper oxidase superfamily member, the nitric oxide reductase (NorB). (c) Synteny diagram of cytochrome ubiquinol oxidase operon genes (*cyoA*, *cyoB*, *cyoC*, *cyoD*, and *cyoE*) with operon from *E. coli* K-12 as a reference. Scale bars correspond to the average number of substitutions per site across alignment.

allocations for soil metagenomics, we were only able to recover genomes for these understudied community members through size fractionating of buffered soil. Our results indicate that CPR bacteria and DPANN archaea are relatively rare in soil, as they can be difficult to recover with typical metagenomic sequencing allotments.

While the precise ecological roles of these organisms remain unclear, their predicted requirement for interaction with nearby community members to satisfy their metabolic needs and their previously reported close physical association with other cells (5, 7) suggest that they may play still undescribed roles in soil microbial interaction networks.

The ability to selectively filter soil solutions to recover CPR and DPANN genomes suggests that either these organisms attach to larger microbial cells and the association can be physically disrupted or they are, at times, not attached to other cells. The approach enabled us to sample genetic inventories of rare soil-adapted microbes and uncover numerous genes and pathways, some of which likely evolved to handle symbiotic lifestyles under relatively oxic conditions. Specifically, we expanded the known diversity of genes and pathways in soil-adapted CPR bacteria and found that these inventories could explain the presence of these organisms, widely understood to be anaerobic, in soil. More generally, our approach provides a route to expand the known diversity of genes and pathways in the soil biosphere.

**Data availability.** Curated genomes described in this study are available from ggKbase (https://ggkbase.berkeley.edu/soilcpr; please note that it is necessary to register for an account by provision of an email address before download) and are available under NCBI BioProject accession number PRJNA744897. NCBI accession numbers for metagenome-assembled genomes are provided in Table S2B.

## SUPPLEMENTAL MATERIAL

Supplemental material is available online only.
**TEXT S1**, DOCX file, 0.1 MB.
**FIG S1**, TIF file, 1.4 MB.
**TABLE S1**, TIF file, 0.6 MB.
**TABLE S2**, XLSX file, 0.1 MB.

## ACKNOWLEDGMENTS

This research was supported by the U.S. Department of Energy Office of Science, Office of Biological and Environmental Research Genomic Science program under awards DE-SC0020163 and DE-SC0016247 to M.K.F. and the NIH grant DP2AI117984 to M.E.T. Work conducted at Lawrence Livermore National Laboratory was supported by the U.S. Department of Energy Office of Science, Office of Biological and Environmental Research Genomic Science program under award SCW1678, LLNL Lab Directed Research and Development award 18-ERD-041, and under the auspices of the U.S. DOE under contract DE-AC52-07NA27344. We acknowledge funding support to J.F.B. from the Chan Zuckerberg Biohub and the Innovative Genomics Institute at UC Berkeley.

We thank Cindy J. Castelle for support inferring archaea phylogenetic relationships and Katerina Estera-Molina for her expertise and management of the Firestone Lab field work site at the Hopland Research and Extension Center (HREC). We acknowledge that HREC sits on traditional, unceded land of the Pomo Indians.

J.F.B. is a founder of Metagenomi. The other authors declare no competing interests.

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
