## [Reviewer comments · mSystems]

Soil CPR encode components of aerobic metabolism and co-occur with nanoarchaea in the rare biosphere of rhizosphere grassland communities

Alexa Nicolas, Alexander Jaffe, Erin Nuccio, Michiko Taga, Mary Firestone, and Jillian Banfield

Corresponding Author(s): Jillian Banfield, University of California, Berkeley

Review Timeline:

Submission Date:	November 18, 2020
Editorial Decision:	February 5, 2021
Revision Received:	April 2, 2021
Editorial Decision:	May 5, 2021
Revision Received:	July 8, 2021
Accepted:	July 31, 2021

Editor: Haiyan Chu

Reviewer(s): Disclosure of reviewer identity is with reference to reviewer comments included in decision letter(s). The following individuals involved in review of your submission have agreed to reveal their identity: Kiel Hards (Reviewer #4)

Transaction Report:

DOI: <https://doi.org/10.1128/mSystems.01205-20>

February 5, 2021

Dr. Jillian F. Banfield
University of California, Berkeley
BERKELEY, CA

Re: mSystems01205-20 (**Soil CPR encode components of aerobic metabolism and co-occur with nanoarchaea in the rare biosphere of rhizosphere grassland communities**)

Dear Dr. Jillian F. Banfield:

Below you will find the comments of the reviewers.

To submit your modified manuscript, log onto the eJP submission site at <https://msystems.msubmit.net/cgi-bin/main.plex>. If you cannot remember your password, click the "Can't remember your password?" link and follow the instructions on the screen. Go to Author Tasks and click the appropriate manuscript title to begin the resubmission process. The information that you entered when you first submitted the paper will be displayed. Please update the information as necessary. Provide (1) point-by-point responses to the issues raised by the reviewers as file type "Response to Reviewers," not in your cover letter, and (2) a PDF file that indicates the changes from the original submission (by highlighting or underlining the changes) as file type "Marked Up Manuscript - For Review Only."

Due to the SARS-CoV-2 pandemic, our typical 60 day deadline for revisions will not be applied. I hope that you will be able to submit a revised manuscript soon, but want to reassure you that the journal will be flexible in terms of timing, particularly if experimental revisions are needed. When you are ready to resubmit, please know that our staff and Editors are working remotely and handling submissions without delay. If you do not wish to modify the manuscript and prefer to submit it to another journal, please notify me of your decision immediately so that the manuscript may be formally withdrawn from consideration by mSystems.

Sincerely,

Haiyan Chu

Editor, mSystems

Journals Department
Reviewer comments:

Reviewer #2 (Comments for the Author):

I read with interest the manuscript entitled "Soil CPR encode components of aerobic metabolism and co-occur with nanoarchaea in the rare biosphere of rhizosphere grassland communities"

This paper examines the diversity of CPR bacteria and DPANN archaea from soil rare biosphere by size-fractionating small particles. The authors found that these microorganisms have aerobic capacity acquired via lateral transfer that may enable adaptation to oxic soil environments. This study provides an approach to understand the soil rare biosphere and its genetic potential. I think this paper is well organized and the nanoparticle concentrate may help to explore unknown microorganisms in the soil. There are some minor issues need to be revised (see below).

1. In the supplementary methods, the soil sampling process (ensure no impurity pollution) and the selection of six field plots should be described.
2. In the methods, DNase was used to remove extracellular DNA that could have derived from larger lysed cells, how do we ensure that DNase do not degrade DNA from small cells and particles in this process?

Comments to authors

Manuscript: Soil CPR encode components of aerobic metabolism and co-occur with nanoarchaea in the rare biosphere of rhizosphere grassland communities

General comments:

The authors pose a highly relevant paper in context of soil microbial ecology that focuses on exploring the rare biosphere of soil particularly the understudied and rare CPR bacteria and DPANN archaea. The manuscript depicts the use of a size fractionation strategy to get enriched CPR and DPANN taxa in soil. The filtered and concentrated DNA samples are compared to bulk soil DNA extraction samples while subjecting both sets to shotgun sequencing. Results show that recovery of genomes of understudied soil community members by fractionation was more robust compared to direct extraction methods. Furthermore, other relevant information as to lateral gene transfers from Proteobacteria to CPR bacteria and evidence of CPR bacteria and DPANN archaea having metabolic strategies evolved to thrive symbiotically in relatively oxic conditions in soil are represented.

Dependency on other cells for CPR lineages have been described. However, it would be nice to have some relevance to the recovery of so many CPR lineages from rhizosphere soil. Furthermore, what motivated the collection from rhizosphere soil to test the hypothesis (greater microbial diversity?); are there background studies that could be referenced in the main manuscript in this regard? Why was the filtration to small fractions done using the rhizosphere soil but not bulk soil (non-plant associated); is the comparison here to see the difference between fractionated and non-fractionated metagenomes or rhizosphere and bulk soil? I think the terminology used such as bulk soil and rhizosphere soil needs clarification because ideally bulk soil refers to outside rhizosphere area whereas rhizosphere is plant associated. I think it is being used here to refer to only rhizosphere soil but which is treated differently in 2 ways as shown in Figure 1.

Overall, I have minor editorial recommendations, see below:

Specific comments:

Line 81-82- Field management information, plant species from where the rhizosphere soil was collected are relevant information that could be included in the details of the main manuscript. Is there a reason why grassland was chosen? How would the results look like in agricultural soils with cropping practices/ actively farmed land?

Line 46- The word lateral gene transfer or horizontal gene transfer might be a better fit

Line 86-88- “bulk DNA was extracted from the same soil samples” please explain the difference between the rhizosphere and bulk soil samples. For example: soil attached to roots were collected as rhizosphere and loosely attached soil was considered as bulk soil, if that was the case. Technically, bulk and rhizosphere are not the same physical sample, so please elaborate briefly as it may cause confusion to the reader. Or as per Fig 1, is it the same soil but treated different (in that case, I suggest refraining from using the word: bulk soil to avoid confusion).

Line 101: specify rhizosphere bacterial sequences to be more clear

Line 144: can a specific link to rhizosphere environments be made here versus just surface soil?
What would the enrichment of CPR lineages mean for the plant, given the focus is on rhizosphere soil here?

Response to Reviewer #2:

This paper examines the diversity of CPR bacteria and DPANN archaea from soil rare biosphere by size-fractionating small particles. The authors found that these microorganisms have aerobic capacity acquired via lateral transfer that may enable adaptation to oxic soil environments. This study provides an approach to understand the soil rare biosphere and its genetic potential. I think this paper is well organized and the nanoparticle concentrate may help to explore unknown microorganisms in the soil. There are some minor issues need to be revised (see below).

1. In the supplementary methods, the soil sampling process (ensure no impurity pollution) and the selection of six field plots should be described.

We have addressed this comment by adding to the supplement the following:

Soil samples were taken from six randomly selected plots on February 16, 2018. These plots belong to a larger ongoing field experiment at the University of California Hopland Research and Extension Center (HREC). The experimental plots at HREC experience a typical Mediterranean climate consisting of cool, wet winters and warm dry summers. Plots contained a variety of naturalized annual grasses and forbs, but were dominated by the naturalized wild oat *Avena barbata*. The top 10 cm of the six randomly selected plots were sampled for soil on February 16, 2018. Soil was collected to ensure no cross contamination amongst the different plots. Samples were homogenized in the field in separate sterile disposable containers and stored on wet ice in whirlpack bags until brought back to the lab for further processing.

2. In the methods, DNase was used to remove extracellular DNA that could have derived from larger lysed cells, how do we ensure that DNase do not degrade DNA from small cells and particles in this process?

That is a good question. Treating environmental samples with DNase prior to extraction of viral and small cell nucleotides is a well established method across systems (Hurwitz et al. 2013; Kleiner et al. 2015) and within soil (Trubl et al. 2018). However, it seems it may be possible that DNase does degrade some cells or particles. In Brinkman et al. 2018, they treat wastewater samples containing an adenovirus spike-in with DNase and find a 46% reduction in adenovirus DNA. Our study shows that we are able to enrich organisms with small cell sizes using the methods described. Perhaps we may have seen increased enrichment ratios compared to non-fractionated metagenomes without the DNase treatment step, but with the tradeoff that we may have also sampled free DNA from soil. This could be an interesting avenue for further research, but was not part of the scope of the study.

May 5, 2021

Prof. Jillian F Banfield
University of California, Berkeley
BERKELEY, CA

Re: mSystems01205-20R1 (**Soil CPR encode components of aerobic metabolism and co-occur with nanoarchaea in the rare biosphere of rhizosphere grassland communities**)

Dear Prof. Jillian F Banfield:

Thank you for submitting your manuscript to mSystems. We have completed our review and I am pleased to inform you that, in principle, we expect to accept it for publication in mSystems. However, acceptance will not be final until you have adequately addressed the reviewer comments.

Thank you for the privilege of reviewing your work. Below you will find instructions from the mSystemseitorial office and comments generated during the review.

Preparing Revision Guidelines

For complete guidelines on revision requirements, please see the Instructions to Authors at [link to page]. **Submissions of a paper that does not conform to mSystems guidelines will delay acceptance of your manuscript.**

Due to the SARS-CoV-2 pandemic, our typical 60 day deadline for revisions will not be applied. I hope that you will be able to submit a revised manuscript soon, but want to reassure you that the journal will be flexible in terms of timing, particularly if experimental revisions are needed. When you are ready to resubmit, please know that our staff and Editors are working remotely and handling submissions without delay. If you do not wish to modify the manuscript and prefer to submit it to another journal, please notify me of your decision immediately so that the manuscript may be formally withdrawn from consideration by mSystems.

Sincerely,

Haiyan Chu

Editor, mSystems

Journals Department
Reviewer comments:

Reviewer #4 (Comments for the Author):

Nicolas et al provide a revised manuscript, using a novel size-based approach to reconstruct several draft genomes from the poorly understood CPR and DPANN phyla. The potential for some of these organisms to possess aerobic respiration capability is highlighted and is an interesting justification for future research. I find the responses to reviewer 2 to be acceptable. I have a further minor issues for the authors to clarify or revise.

These comments are mainly aimed around the author's proposal that Saccharibacteria, Parcubacteria and Doudnabacteria perform aerobic respiration. Particularly, the authors may wish to further consult literature, to comment on the activity of an isolated bo3 complex in absence of other electron transport chain components.

1) Line 134: Is there any literature evidence that cytochrome bo3 can detoxify reactive oxygen species? To my knowledge this ability is a hallmark of the bd family of oxidases, not bo3 (doi: 10.1016/j.febslet.2013.05.047), so literature one way or the other may help the authors better speculate on the role of bo3. Does the co-occurrence of F-type ATP synthases with cytochrome bo3 (Fig 1D) suggest that the role of cytochrome bo3 in these bacteria is energy conservation? The authors may wish to include considerations around the ATP synthases in their discussion.

2) What is the expected source of reductant for cytochrome bo3 activity? Have the authors identified any quinone biosynthesis genes, or searched for alternative electron donating dehydrogenases that are less well annotated by KO (e.g. [NiFe]-Hydrogenases, Type II NADH dehydrogenases, Malate:Quinone Oxidoreductases)? If the authors are confident that the

cytochrome bo3 exists essentially as an "orphan" respiratory complex, then the authors may wish to consider the possibility of interspecies electron transfer (reviewed in doi: 10.1146/annurev-micro-030117-020420), using quinone (e.g. with a *Shewanella* donor as per doi: 10.7554/eLife.48054.001) given the variety of symbiotic interactions for these organisms.

3) To further support the likelihood of horizontal transfer of bo3: Are there any obvious difference in AT/GC content in the bo3 operons compared to the rest of the draft genomes?

4) Line 129: Have the authors tried any approaches beyond looking at the annotation of the oxidoreductase and conserved hypothetical protein? Many studied bacterial ETC complexes are poorly annotated.

Minor Comments:

- CPR is not defined in the title
- DPANN would be better defined as just the phyla names in the abstract
- Minor reference list formatting issues e.g. inconsistent use of doi in bioRxiv references
- Figure 1: The authors should define genes that they don't discuss in text (e.g. PPA, sdhA)

Response to Reviewer #4

We appreciate the additional comments provided by reviewer #4. We believe this feedback is invaluable to make the final manuscript much stronger, more precise, and thoughtful. We have addressed the following comments below:

1) Line 134: Is there any literature evidence that cytochrome bo3 can detoxify reactive oxygen species? To my knowledge this ability is a hallmark of the bd family of oxidases, not bo3 (doi: 10.1016/j.febslet.2013.05.047), so literature one way or the other may help the authors better speculate on the role of bo3. Does the co-occurrence of F-type ATP synthases with cytochrome bo3 (Fig 1D) suggest that the role of cytochrome bo3 in these bacteria is energy conservation? The authors may wish to include considerations around the ATP synthases in their discussion.

After consultation with literature, we agree that cytochrome bo3 is unlikely to detoxify reactive oxygen species and have removed these mentions from the manuscript accordingly. Regarding the co-occurrence of the F-type ATP synthases with the cytochrome bo3, this is an interesting point that we have added to the manuscript text as potential evidence of energy conservation in these genomes.

2) What is the expected source of reductant for cytochrome bo3 activity? Have the authors identified any quinone biosynthesis genes, or searched for alternative electron donating dehydrogenases that are less well annotated by KO (e.g. [NiFe]-Hydrogenases, Type II NADH dehydrogenases, Malate:Quinone Oxidoreductases)? If the authors are confident that the cytochrome bo3 exists essentially as an "orphan" respiratory complex, then the authors may wish to consider the possibility of interspecies electron transfer (reviewed in doi: 10.1146/annurev-micro-030117-020420), using quinone (e.g. with a Shewanella donor as per doi: 10.7554/eLife.48054.001) given the variety of symbiotic interactions for these organisms.

We did not identify any quinone biosynthesis genes and have added this to the text. We did identify an NADH dehydrogenase (K03885; EC:1.6.99.3) that often co-occurs with the cytochrome bo3 and have added this as a new column in Figure 1D and in the manuscript text. Taken together, these comments help strengthen the idea that these organisms may be capable of some aerobic metabolism. We very much appreciate these thoughtful comments.

3) To further support the likelihood or horizontal transfer of bo3: Are there any obvious difference in AT/GC content in the bo3 operons compared to the rest of the draft genomes?

This is a great suggestion, but given that this is likely not a recent horizontal gene transfer, since there seems to be vertical inheritance within CPR phyla of the bo3 operon, we would not expect GC content in the bo3 operons to differ significantly from the rest of genomes to serve as a way of evidencing horizontal gene transfer. Further, there is evidence that most horizontal transfers of metabolic genes in CPR are quite ancient (Jaffe et al 2020; DOI: 10.1186/s12915-020-00804-5). Regardless, the inferences based on the maximum likelihood tree are the most robust way of ascertaining whether there is evidence of horizontal gene transfer.

4) Line 129: *Have the authors tried any approaches beyond looking at the annotation of the oxidoreductase and conserved hypothetical protein? Many studied bacterial ETC complexes are poorly annotated.*

We agree that this is an interesting direction for analysis, we have tried BLASTing these sequences, using tools to predict potential functional domains (e.g. HHPred), but further interrogation of these sequences fell outside the scope of this observation-lengthened study. We hope future research looks into whether these are truly functional cytochrome operons and tries to identify these poorly characterized ORFs associated with the operon, CPR biology contains some very exciting chemistry.

Minor Comments:

- CPR is not defined in the title

Term defined in title

- DPANN would be better defined as just the phyla names in the abstract

Updated to: DPANN (archaea of the phyla Diapherotrites, Parvarchaeota, Aenigmarchaeota, Nanoarchaeota and Nanohaloarchaeota)

- Minor reference list formatting issues e.g. inconsistent use of doi in bioRxiv references

References updated.

- Figure 1: The authors should define genes that they don't discuss in text (e.g. PPA, sdhA)

Definitions were added to the Figure 1 legend.

July 31, 2021

Prof. Jillian F Banfield
University of California, Berkeley
BERKELEY, CA

Re: mSystems01205-20R2 (**Soil CPR encode components of aerobic metabolism and co-occur with nanoarchaea in the rare biosphere of rhizosphere grassland communities**)

Dear Prof. Jillian F Banfield:

Your manuscript has been accepted, and I am forwarding it to the ASM Journals Department for publication. For your reference, ASM Journals' address is given below. Before it can be scheduled for publication, your manuscript will be checked by the mSystems senior production editor, Ellie Ghatineh, to make sure that all elements meet the technical requirements for publication. She will contact you if anything needs to be revised before copyediting and production can begin. Otherwise, you will be notified when your proofs are ready to be viewed.

As an open-access publication, mSystems receives no financial support from paid subscriptions and depends on authors' prompt payment of publication fees as soon as their articles are accepted. =

Publication Fees:

We recognize that the video files can become quite large, and so to avoid quality loss ASM

suggests sending the video file via <https://www.wetransfer.com/>. When you have a final version of the video and the still ready to share, please send it to Ellie Ghatineh at eghatineh@asmusa.org.

Sincerely,

Haiyan Chu
Editor, mSystems

Journals Department
Supplemental Material: Accept
Supplemental Material: Accept
Supplementary Methods and Materials: Accept
Supplemental Material: Accept